# Valorisation of Side Stream Products through Green Approaches: The Rapeseed Meal Case

**DOI:** 10.3390/foods12173286

**Published:** 2023-09-01

**Authors:** Francesco Cairone, Dario Allevi, Stefania Cesa, Giancarlo Fabrizi, Antonella Goggiamani, Domiziana Masci, Antonia Iazzetti

**Affiliations:** 1Dipartimento di Chimica e Tecnologie del Farmaco, Sapienza, Università di Roma, P.le A. Moro 5, 00185 Rome, Italy; francesco.cairone@uniroma1.it (F.C.); stefania.cesa@uniroma1.it (S.C.); giancarlo.fabrizi@uniroma1.it (G.F.); antonella.goggiamani@uniroma1.it (A.G.); 2Dipartimento di Scienze Biotecnologiche di Base, Cliniche Intensivologiche e Perioperatorie, Università Cattolica del Sacro Cuore, L.go Francesco Vito 1, 00168 Rome, Italy; dario.allevi@unicatt.it (D.A.); domiziana.masci@unicatt.it (D.M.); 3Policlinico Universitario ‘A. Gemelli’ Foundation-IRCCS, 00168 Rome, Italy

**Keywords:** rapeseed meal, sinapic acid, supercritical fluid extraction, polyphenols, green methods

## Abstract

Rapeseed meal (RSM) is a by-product of rapeseed oil extraction and is a rich source of bioactive compounds, including proteins and antioxidants. This study compared two methods for extracting antioxidants from RSM: conventional ethanol Soxhlet extraction and supercritical CO_2_ extraction. These procedures were applied to both native RSM and RSM after protein removal to evaluate their bio-compound composition and potential applications. HPLC-DAD, NMR, and GC/MS analyses revealed a rich polyphenolic profile in the extracts, including the presence of sinapic acid. The concentration of sinapic acid varied depending on the extraction method used. The anti-radical activity of the extracts was also analysed using the DPPH assay, which confirmed the potential of RSM as a source of antioxidants for use in cosmetics, food, and pharmaceutical formulations.

## 1. Introduction

In 2020, rapeseed was the second most cultivated oilseed crop in the world, accounting for 68 million tons, with France, Germany, and Poland being the main producing countries [1]. As a result, rapeseed meal (RSM), obtained after oil extraction, is also produced in large quantities worldwide (40 million tonnes/year) and its production reached 12.5 million tons in the EU in 2020. In particular, in Europe, RSM is an important feed for the production of biodiesel, so the transformation of biomass resulting from its processing into materials and energy with high added value has become a powerful tool to increase the competitiveness and sustainability of biofuels [2].

RSM is considered a rich protein source, with a global output of 73 million tons in 2017, second best only to soybean products [3]. As a suitable protein source, it represents a potential energy source in animal feed. Moreover, approximately 35% of RSM dry matter is composed of carbohydrates, half of which are water-soluble carbohydrates, such as arabinan, galactomannan, homogalacturonan, rhamnogalacturonan I, type II arabinogalactan, glucuronoxylan, and cellulose. RSM also presents a rich phytocomplex characterised by interesting minor bio-components such as polyphenols and tocopherols [4,5] with antioxidant activity.

Indeed, rapeseed contains high levels of phenolic compounds, including phenolic acids and condensed tannins. These compounds are more abundant in rapeseed products than in products made from other oilseeds [4]. Phenolic compounds such as free phenolic acids, sinapines, and condensed tannins can contribute to the bitter taste and astringency of rapeseed products. They can also form complexes with proteins, reducing the nutritional value of the products. To extract these phenolic compounds, various extraction procedures have been studied. These secondary metabolites and plant phenolic compounds are stored in the outer layers of oilseeds and can be difficult to extract due to their association with lipoprotein bilayers and the cell wall [5]. Among the most representative polyphenolic compounds found in rapeseed are vanillic, ferulic, p-coumaric, chlorogenic, caffeic, and sinapic acid [6]. In particular, sinapic acid and its derivatives (i.g., sinapin) are the main phenolic acids present in RSM (about 70%). Therefore, whereas rapeseed oil is mainly consumed in the human diet, rapeseed meal is a co-product commonly used as a protein source in animal diets. Since the application of RSM in nutrition is limited due to the presence of anti-nutritional factors and toxic substances, [2,3,4,5] many technologies have been developed to improve nutrient digestibility aiming at more efficient pig growth [6,7].

As a raw material, RSM presents a protein content between 35% and 40%, which mainly constitutes arginine, histidine, leucine, lysine, threonine, and valine [7]. The high content of essential amino acids and, particularly, lysine could be the best parameter that indicates the protein quality in RSM [8]. Additionally, as RSM represents the by-product of rapeseed oil extracted via different processing techniques, it can be characterised by a very different residue of crude fat, ranging between 2 and 15% in content. This content, in fact, depends on the type of rapeseed, the impurity content, the processing technology, and other factors. More specifically, significant qualitative differences in fat content were observed. After storage, the content of butyric, caproic, caprylic, and palmitoleic acids increased, while the concentration of lauric, stearic, and oleic acids decreased. The proportion of saturated fatty acids in total fatty acids increased (19.7% vs. 14.3%) and that of unsaturated fatty acids declined (80.3% vs. 85.7%) [9]. Moreover, RSM is a relatively rich source of minerals, including calcium, phosphorus, potassium, iron, zinc, and selenium [8,9].

Although RSM is mainly exploited for its amino acid content as a valuable protein source, it is also a rich source of phenolic compounds with nutritional bioavailability and fibres with sensory and functional properties. These molecules deserve to be valorised for their high antioxidant, antimicrobial and health-promoting activities [10]. From this point of view, RSM, could be recycled as a source of nutrients and bioactive molecules to use in the development of nutraceuticals, cosmetics, and pharmaceuticals, as well as in food and feed sectors.

The recovery of phenolic compounds from rapeseed can be strongly influenced by processing conditions such as an elevated temperature and pressure. Indeed, these parameters can affect the solubility and stability of the phenolic compounds, as well as their interactions with other components in rapeseed. As a result, it is important to carefully control the processing conditions in order to optimise the extraction of phenolic compounds from rapeseed [11].

The aim of the present work was to study rapeseed meal composition and valorise the bioactive molecule composition through innovative and “green” extraction procedures such as supercritical CO_2_ extraction [12,13,14,15]. Great attention was paid to the better recovery of specific polyphenolic compounds, the presence of which was highlighted by means of NMR, GC/MS, and HPLC-DAD analyses. These analytical techniques, widely used in the analysis of phenolic compounds in foods, nutraceuticals and medicinal plants, can provide detailed information on the composition of extracts and are complementary to the identification of specific classes of compounds [12,13].

## 2. Materials and Methods

### 2.1. Chemicals

Ethanol and other HPLC-grade solvents, and standard compounds were purchased from Merck Science Life (Milan, Italy). Deuterated solvents were purchased from Eurisotop (Saint-Aubin, France). NaOH pellets and a 37% HCl solution were provided by Carlo Erba reagents (Cornaredo, Italy). The CO_2_ gas tank was purchased from Sapio (Monza, Italy).

### 2.2. Plant Materials

Rapeseed meal, obtained from a biorefinery plant (ENVIRAL a.s., Leopoldov, Slovak Republic), was homogenised with a mixer (Bimby^®^ Vormek TM6, Germany, Italy) before use (**R1**).

### 2.3. Protein Removal

Protein removal has been performed in accordance with a slightly modified procedure in the literature [12].

About 10 g of ground rapeseed meal (**R1**) was suspended in NaOH 2 N (100 mL), and was left to be stirred for 16 h at room temperature (25 °C). The resulting suspension was centrifuged (2000× *g*, 10′), while the supernatant’s aqueous layer and the protein-rich precipitate (**R2**) were collected and stored at 4 °C. The aqueous layer was acidified dropwise using 2 N HCl at 5 °C to precipitate the proteins that were removed for centrifugation. The aqueous layer was then neutralised before to be lyophilised (**R3**) and stored at 4 °C. The proposed workflow is outlined in Figure 1.

### 2.4. Ethanol Extraction

About 5 g of **R1**, **R2**, and **R3** was extracted with 50 mL of ethanol, with a Soxhlet apparatus for 2 h. The resultant ethanolic solutions were dried over a vacuum at 40 °C and stored at 4 °C until subsequent analysis (**R1a**, **R2a**, and **R3a**).

### 2.5. SFE-CO_2_ Extraction

About 5 g of **R1**, **R2**, and **R3** samples was subjected to CO_2_ supercritical fluid extraction (SFE-CO_2_). SFE-CO_2_ extraction was performed using a supercritical CO_2_ apparatus provided by Jasco Europe (Cremella, Italy). The system consists of a pump (scCO_2_ Jasco-PU-4347), a controlled temperature extraction oven with a stainless steel filtering set (Jasco CO-4065), and a back-pressure regulator (Jasco CO_2_-BP-4390). Extraction was thus performed using ethanol as a co-solvent (CO_2_/EtOH 90/10 *v*/*v*), at a flow rate of 5 mL/min, for 2 h at 40 °C and at 10 MPa. The resultant ethanolic solutions were dried under a vacuum at 40 °C and stored at 4 °C until subsequent analyses (**R1b**, **R2b**, and **R3b**).

### 2.6. Transesterification Step and GC/MS Analysis

Samples **R1a**, **R1b** and **R2a** were subjected to a transesterification procedure in accordance with a procedure previously reported in the literature [13]. In brief, about 100.0 mg of the samples was dissolved in 0.2 mL of MeOH. Then, MeONa was added (24.1 mg, 0.9 mmol) and the mixture was stirred for 3 h at room temperature. After this time, the crude compound was diluted with AcOEt (25 mL) and washed with water (3 × 10 mL). The combined organic layers were dried over Na_2_SO_4_, filtered, concentrated under a vacuum and then were subjected to GC/MS analyses. GC/MS analyses were performed with QP-2010-Plus Gas Chromatograph Mass Spectrometer (Shimadzu Italia S. r. l., Milan, Italy) using a “silica fusa” Rix^®^-5ms column (Restel^®^) (30 m, 0.25 mm ID, 0.25 µm) (Restel S.r.l., Cernusco Sul Naviglio, Milan, Italy), with a flow of 1.0 mL/min. The temperature varied from 100 °C to 250 °C in 10′.

### 2.7. ^1^H NMR Analysis

Samples (in amounts of about 10–15 mg) were dissolved in DMSO-d_6_ (0.7 mL), CDCl_3_ (0.7 mL) or MeOD (0.7 mL) and poured into an NMR analysis tube. ^1^H-NMR (400.13 MHz) and ^13^C-NMR (100.6 MHz) analyses were performed using Bruker Avance 400 (Milan, Italy), equipped with a Nanobay console and Cryoprobe Prodigy probe.

### 2.8. HPLC Analysis

The extracted samples were weighed and dissolved in HPLC-grade methanol. The obtained solutions (5 mg/mL) were filtered with a Millex^®^ LG filter (Low Protein Binding Hydrophilic PTFE 0.20 µM Membrane) (Merck Science Life, S.r.l., Milan, Italy) and then injected into an HPLC-DAD instrument purchased from Perkin Elmer (Milan, Italy). The chromatographic analyses were carried out at 280 nm (for the identification of hydroxycinnamic acids) and 360 nm (for the identification of flavonoids), using a Luna RP-18 3 μ chromatographic column, at room temperature. The mobile phase consisted of acetonitrile (A) and a formic acid solution at a concentration of 5% in water (B). The binary gradient used was as follows: from 0% A—100% B to 60% A–40% B in 45 min. The flow was 0.8 mL/min. The identification and the quantification of the molecules of interest such as gallic acid (y = 15.51x + 37.06; R^2^ = 0.9987, in the range between 3 and 200 µg/mL, at a LOD of 0.09 µg/g and LOQ of 0.3 µg/g of the extract in dry weight), chlorogenic acid (y = 12.02x − 3.95; R^2^ = 0.9991, in the range between 8 and 160 µg/mL, at a LOD of 0.07 µg/g and LOQ of 0.2 µg/g of the extract in dry weight), caffeic acid (y = 35.23x − 28.86; R^2^ = 0.9989, in the range between 4 and 160 µg/mL, at a LOD of 0.05 µg/g and LOQ of 0.2 µg/g of the extract in dry weight), sinapic acid (y = 11.37x + 9.92; R^2^ = 0.9987, in the range between 2 and 100 µg/mL, at a LOD of 0.9 µg/g and LOQ of 3 µg/g of the extract in dry weight) and rutin (y = 13.60x + 33.11; R^2^ = 0.9994, in the range between 2 and 200 µg/mL, at a LOD of 0.06 µg/g and LOQ of 0.2 µg/g of the extract in dry weight) were carried out using external standard calibration curves. The detailed protocol related to HPLC-DAD analysis is described in the Appendix A.

### 2.9. DPPH Assay

The assay was performed following the literature [14]. In brief, 0.5 mL of isopropanol was added to 2.5 mL of a 168 μM solution of DPPH in isopropanol. Solutions were stored in darkness and periodically checked with a UV/VIS spectrophotometer, Lambda25 (Perkin Elmer, Waltham, MA, USA), at 515 nm to monitor radical stability. Then, to 2.5 mL of the same DPPH solution, 0.5 mL of an extract solution (0.05 mg/mL) in isopropanol was added. The resulting absorbance was monitored at 515 nm. The antioxidant activity of the extracts was then calculated, as gallic acid equivalents, using the calibration curve obtained in accordance with the literature (y = 0.6473e^−378.5x^; R^2^ = 0.9994) [16].

### 2.10. Statistical Analysis

Each assay was replicated at least three times. Data are expressed as mean ± sd and statistical significance was determined using the XLStat software (version XLSTAT 2021, New York, NY, USA).

## 3. Results and Discussion

### 3.1. Extraction Yields

As reported, RSM is currently used as a source of proteins in animal feed. It has also been studied as a source of amino acids for diverse applications [2,7,9] with a lack of studies focusing on its antioxidant components. As part of our interest in the valorisation of industrial byproducts [15,17] and in to the pursuit of fully reusing such a precious waste, we decided to focus on its antioxidant content. To start our investigation, we evaluated native RSM (**R1**) as a source of sinapic acid, and then we focused on the residues after protein removal (**R2** and **R3**). pH control in the process of alkaline extraction and acid precipitation was an effective strategy with which to isolate proteins from industrial defatted meal. Alkaline extraction at pH 9.0 and subsequent acid precipitation at pH 4.5 were found to be the best process parameters with which to obtain structurally intact, safe and bioavailable rapeseed protein with a maximum yield and lower amount of anti-nutritional compounds [18]. This procedure allowed the purification of the crude compound from the protein, making the antioxidant compounds available to subsequent extraction steps. Conventional extraction procedures with organic solvents have been applied to the extraction of antioxidant compounds from canola. In particular, different Soxhlet extraction procedures using 80% (*v*/*v*) methanol in a ratio of 1:10 to 1:100 (*w*/*v*), at 50–80 °C for 1–6 h, ethyl acetate and/or n-hexane solvents at 50 °C for 4–6 h are reported [4,19,20]. These methods could have undesirable effects on the environment and on food components; moreover, the use of high temperatures and long extraction times also cause energy efficiency issues. Additionally, in a study by Nandasiri et al., the effect of temperature (140, 160, and 180 °C) and pressure (1.500 psi) on the extraction and yield of phenolic compounds from canola meal as well as the solvent type (ethanol and methanol) and concentration (30%, 40%, 60%, and 70% *v*/*v*) were evaluated. Hence, to study the relative differences between **R1**, **R2**, and **R3**, extraction with the conventional ethanolic solvent and that with supercritical CO_2_were optimised and compared, Refs. [19,20] with the main purpose being to develop a zero-impact process for the valorisation of all the feedstock, reducing the production of waste as much as possible. The gravimetric data obtained via the two different applied methodologies are reported in Table 1.

In particular, the extraction of RSM at 60–80 °C with alcoholic solvents (especially methanol) in a ratio of 1:100 is reported to provide a recovery of 20–80% of total phenolic compounds, which is particularly composed of sinapic acid [19]. On the other hand, regarding extraction in supercritical CO_2_, no promising extraction methods are reported in the literature. However, interesting results have been reported for supercritical water extraction, achieving a recovery of about 30% of phenolic compounds [20]. Based on this evidence, initially, in our work, an ethanolic extraction procedure was optimised, obtaining variable yields between 9% (**R2**) and 35% (**R3**), which not only correspond to the polyphenolic content (representing about the 5% of the extracted material) but correlated with the higher extraction of sugars and fatty acids contained in the analysed matrix (as further shown via NMR analysis). For this reason, to achieve a better recovery of the polyphenolic component, CO_2_ extraction was optimised, using ethanol as a cosolvent. In fact, despite the lower extraction yields (0.7–4%), a recovery of polyphenols of between 13 and 45% was reached (as further shown via HPLC-DAD and NMR analysis),making it possible for them to be better used as nutraceuticals, cosmetics, or dietary supplements.

### 3.2. ^1^H-NMR Analysis and GC/MS

NMR analyses were performed on the extracts obtained from **R1**, **R2**, and **R3** through the two main methodologies previously described. The presence of antioxidant compounds was observed in all the extracts except for **R1b**.

Particularly, the ^1^H-NMR spectrum of the residue obtained from the ground rapeseed meal through ethanol Soxhlet extraction (**R1a**) showed the characteristic signals of sinapic acid (^1^H NMR (400.13 MHz) (DMSO-d_6_), with δ = selected signals, these being 7.62 (d, 1H, J = 15.9 Hz, Ar-CH=CH), 7.04 (s, 2H, Ar-H), 6.55 (d, 1H, J = 15.9 Hz, Ar-CH=CH), and 3.81 (s, 6H, OCH_3_) [17], which in this case, is very likely in the form of a glycoside (Figure 2A). In addition, traces of unsaturated fatty acids were also evident. These results were confirmed via the GC/MS analysis carried out for **R1a**, after a transesterification step (Appendix A). In particular, sample **R1a** presented fatty acid methyl esters along with sinapic acid in the form of methyl-3-(3,4,5-tri methoxyphenyl)acrylate (Figure 2B).

Conversely, the same analysis carried out for the residue obtained via CO_2_/ethanol 9:1 extraction (**R1b**) revealed only the presence of triglycerides and free fatty acids (Figure 3, panel A).

Analysing **R1b**, after transesterification, via GC/MS, the presence of fatty acid methyl esters was confirmed with a prevalence of C18:1 isomers (Figure 3, panel B).

Probably, the poor extraction of sinapic acid observed with supercritical CO_2_ extraction (SFE-CO_2_) was due to the lack of an alkaline treatment which, in the subsequent cases, made the active compounds easier to extract under the milder extractive conditions typical of SFE-CO_2_.

The ^1^H NMR spectrum of **R2a** showed the main presence of free fatty acids and traces of sinapic acid (Figure 4, panel A).

Even in this case, the data were confirmed via GC/MS analyses carried out after a transesterification step (Appendix A). The data indicated the presence of a high amount of C18:1 fatty acid isomers (Figure 4, panel B), in accordance with the literature [16,19].

Notably, in the ^1^H-NMR spectrum of residue **R2b**, the presence of sinapic acid in larger amounts than that in the previously analysed residue (**R2a**) was detected even if the mixture appeared to be mainly composed of fatty acids (Figure 5).

Switching to the ^1^H NMR analysis of **R3a** and **b**, the NMR analysis revealed great similarity in terms of the composition between the two residues. Particularly, both spectra showed the presence aromatic protons of between 7.33 and 7.59 ppm, which, according to the literature, may indicate the presence of benzoic acid derivatives [19,21] (Figure 6 and Figure 7).

In addition, in **R3b** the typical signals of sinapic acid are evident (Figure 7, showing expansion between 9 and 6 ppm), confirming that, even in this case, extraction using supercritical CO_2_ allows the obtention of residues most enriched in this component.

Results highlighted from the NMR analysis are in line with the data obtained by means of HPLC-DAD and DPPH analyses.

### 3.3. HPLC-DAD Analysis

The different extracts (**R1**, **R2**, and **R3**) were subjected to HPLC-DAD analysis, for the identification of benzoic and hydroxycinnamic acids, and of flavonols. The example chromatograms are reported in Figure 8, Figure 9, Figure 10 and Figure 11. In Table 2, the data related to the quantification of bioactive compounds are reported. As reported in the literature, sinapine, the most abundant molecule in RSM, can be hydrolysed into sinapic acid and other derivatives which represent bioactive compounds, improving the health potential correlated to a food’s nutritional quality, such as the content of vanillic, caffeic, and coumaric acids, and flavonols such as quercetin, rutin, and kaempferol as esters and glucosides [20,22]. As shown in Table 2, sinapic acid was the main extracted bioactive compound. In particular, it was present in an amount of about 20 µg/g of the dried extract in **R2b**, whereas it more than halved in the other samples. Sample **R2b** appeared the richest extract in terms of bioactive compounds compared with **R1** (**a** and **b**) and **R3** (**a** and **b**). In fact, in addition to sinapic acid, other molecules were identified such as benzoic acid, identified only in the **R2b** and **R3** (**a** and **b**) samples (expressed as gallic acid equivalents), chlorogenic acid, caffeic acid, identified only in **R2b**, and different flavonols which were expressed as rutin equivalents. These data confirm that the phytocomplex can be better extracted if extraction is performed after protein precipitation. Large differences between the ethanolic and CO_2_ extract were not observed, except in **R1b**. In fact, **R1b** did not show a polyphenolic profile, as confirmed via NMR analyses, which mainly highlighted the presence of free fatty acids. Probably, CO_2_ extraction in the ground rapeseed meal does not perform well in terms of extracting polyphenolic compounds. No particular differences are noted between samples **R3a** and **R3b**. Considering the CO_2_ extraction yields (1–4% vs. 10–35%), there was approximately an eight-fold concentration of the latter.

The data obtained are in line with what has been reported in the literature, where sinapic acid values between 10 and 20 mg/kg are reported and a total phenolics content of about 20–30 mg/kg dried extract are found [23,24,25]. These data are confirmed via NMR analysis.

### 3.4. DPPH Analysis

The DPPH analysis of the obtained CO_2_ and ethanolic extracts (**R1a**, **R1b**, **R2a**, **R2b**, **R3a** and **R3b**) presented a wide range of values between 0.8 and 6 mg/g of gallic acid equivalents (Table 3), confirming the presence of different antioxidant capacities among the samples, according to their different polyphenolic composition, with **R2b** representing the sample with the highest anti-radical activity. Polyphenolic compounds not only act as antioxidant compounds but also prevent lipid peroxidation, influencing the antioxidant activity of the extracts [26]. In fact, in agreement with the NMR and HPLC analyses, the sample showed a high sinapic acid content, as well as a high content of several polyphenolic compounds (see Section 3.3).

In general, the extracts obtained from CO_2_ extraction presented higher antioxidant activity, except for **R1b**, which showed the lowest anti-radical activity (see Section 3.2 and Section 3.3). The results obtained also overlap with what has been reported in other works performed on different food matrices and analysed in our laboratories, such as kiwi, Sulmona red garlic, sour cherry, etc., functional foods considered to have high health potential [10,22,23,24] (Figure 12).

## 4. Conclusions

This study highlights the potential of rapeseed meal as a candidate feedstock for further valorisation through subsequent biorefining processes. An environmentally sustainable and innovative CO_2_ extraction procedure has been optimised and validated for the extraction of bioactive components, proving to be the best procedure in terms of yields and selectivity. The obtained data showed a rich polyphenolic profile, such as that of phenolic acids and flavanols. The **R2b** sample appears to be the most promising in terms of antioxidant content and antiradical activity. The presence of polyphenols, and in particular of sinapic acid, indicates high potential for applications in the nutraceutical and cosmetic sectors and make the development of specific extraction procedures for high-value-added compounds promising tools in the development of a zero-impact circular economy approach.

## Figures and Tables

**Figure 1 foods-12-03286-f001:**
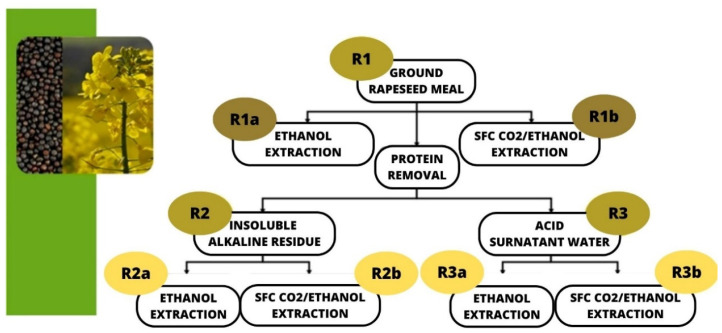
Workflow.

**Figure 2 foods-12-03286-f002:**
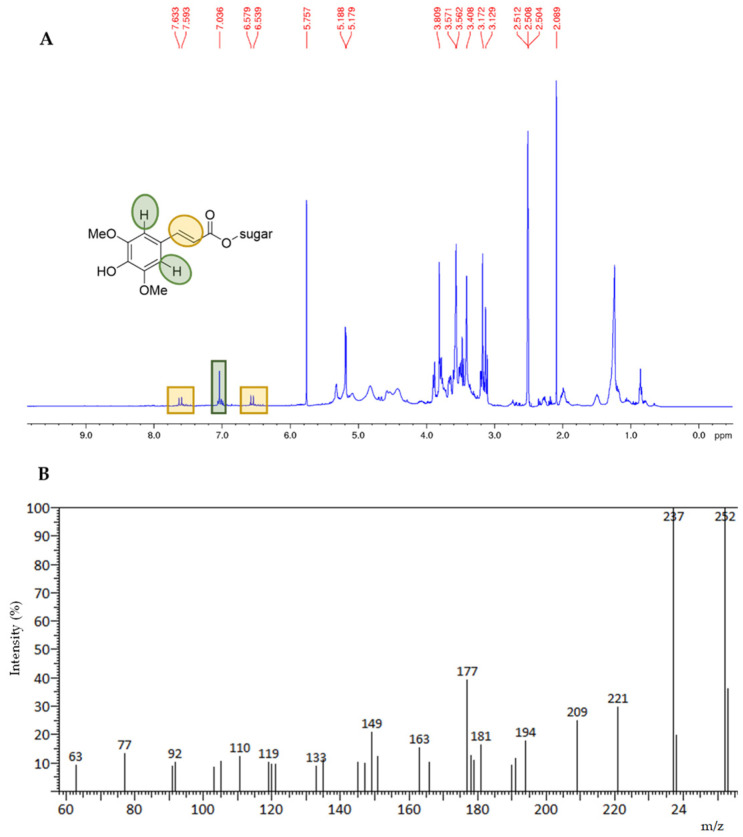
^1^H NMR spectrum of residue **R1a** in DMSO d6 (Panel (**A**)); GC-Mass spectrum of C_13_H_16_O_5_ (M+, m/z 252) {methyl (E)-3-(3,4,5-trimethoxyphenyl)acrylate}, **R1a** residue (Panel (**B**)).

**Figure 3 foods-12-03286-f003:**
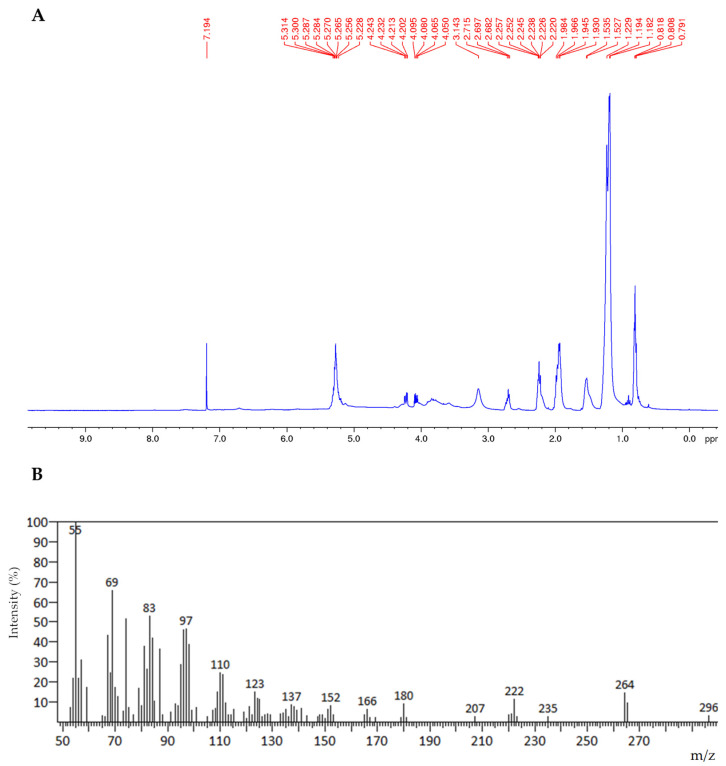
^1^H NMR spectrum of residue **R1b** in CDCl_3_ (Panel (**A**)); GC-Mass spectrum of the methyl esters of C18:1 (M+, m/z 296), **R1b** residue (Panel (**B**)).

**Figure 4 foods-12-03286-f004:**
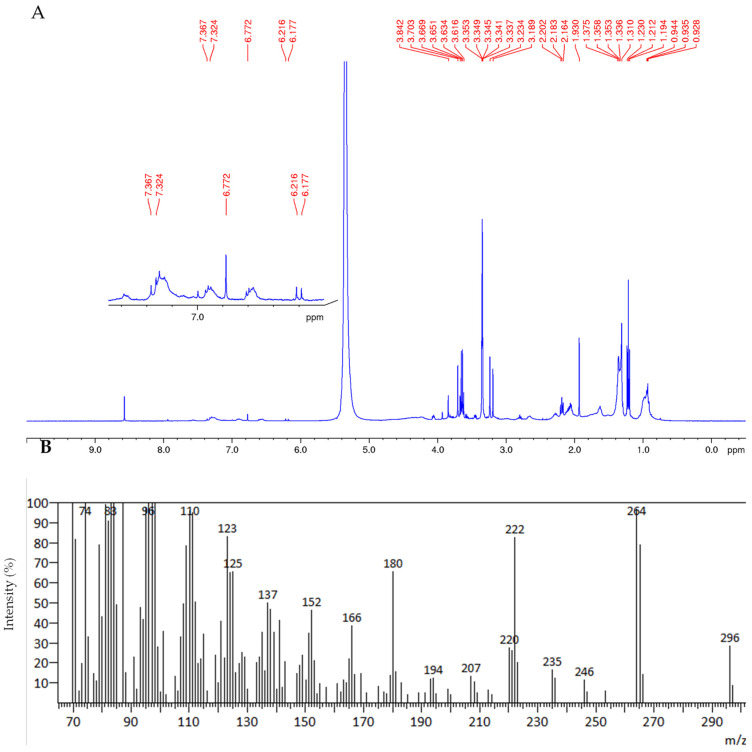
^1^H NMR analysis of residue **R2a** in CD_3_OD (Panel (**A**)); GC-Mass spectrum of the methyl esters of C18:1 (M+, m/z 296), **R2a** residue (Panel (**B**)).

**Figure 5 foods-12-03286-f005:**
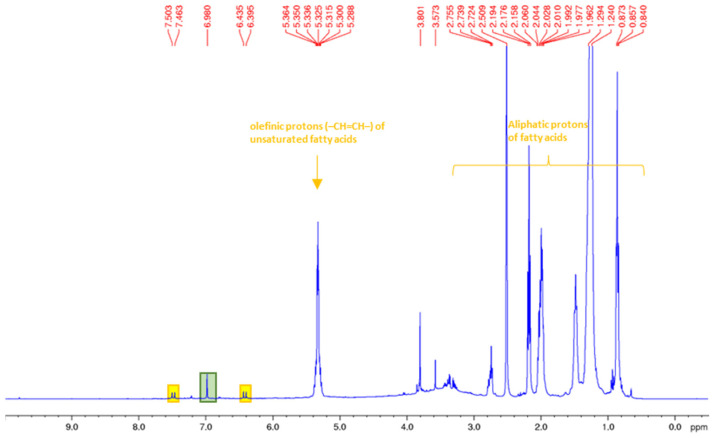
^1^H NMR analysis of residue **R2b** in DMSO d_6_.

**Figure 6 foods-12-03286-f006:**
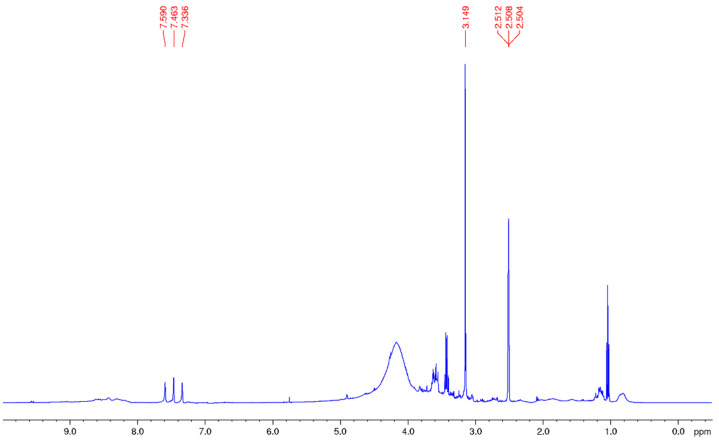
^1^H NMR analysis of residue **R3a** in DMSO d_6_.

**Figure 7 foods-12-03286-f007:**
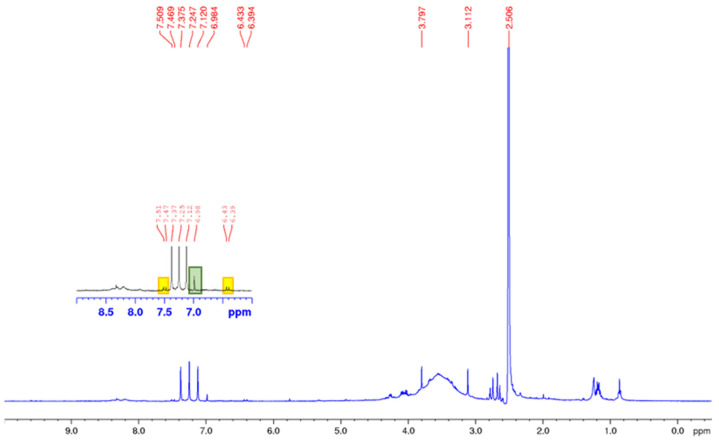
^1^H NMR analysis of residue **R3b** in DMSO d_6_.

**Figure 8 foods-12-03286-f008:**
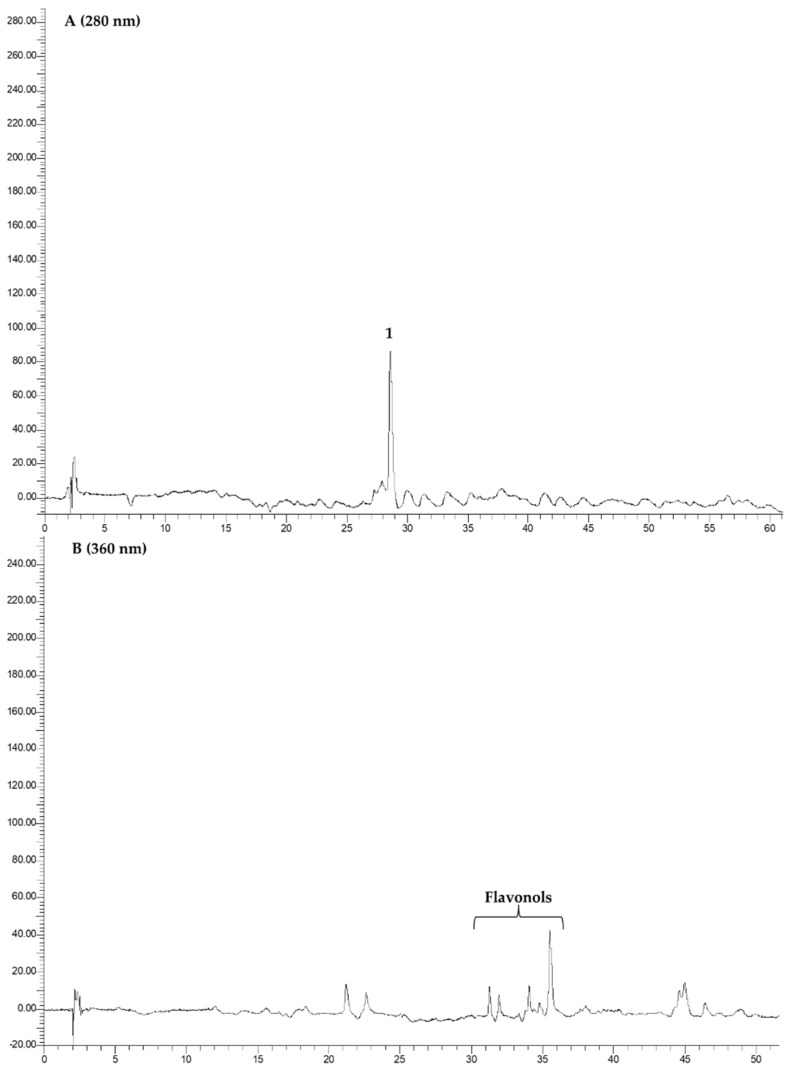
Chromatograms of **R1a** recorded at 280 nm (Panel (**A**)) and at 360 nm (Panel (**B**)). 1. Sinapic acid.

**Figure 9 foods-12-03286-f009:**
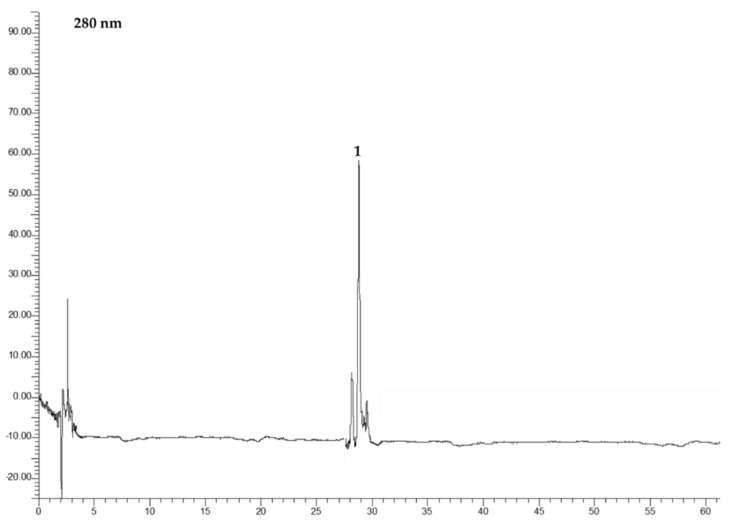
Chromatograms of **R2a** recorded at 280 nm. 1. Sinapic acid.

**Figure 10 foods-12-03286-f010:**
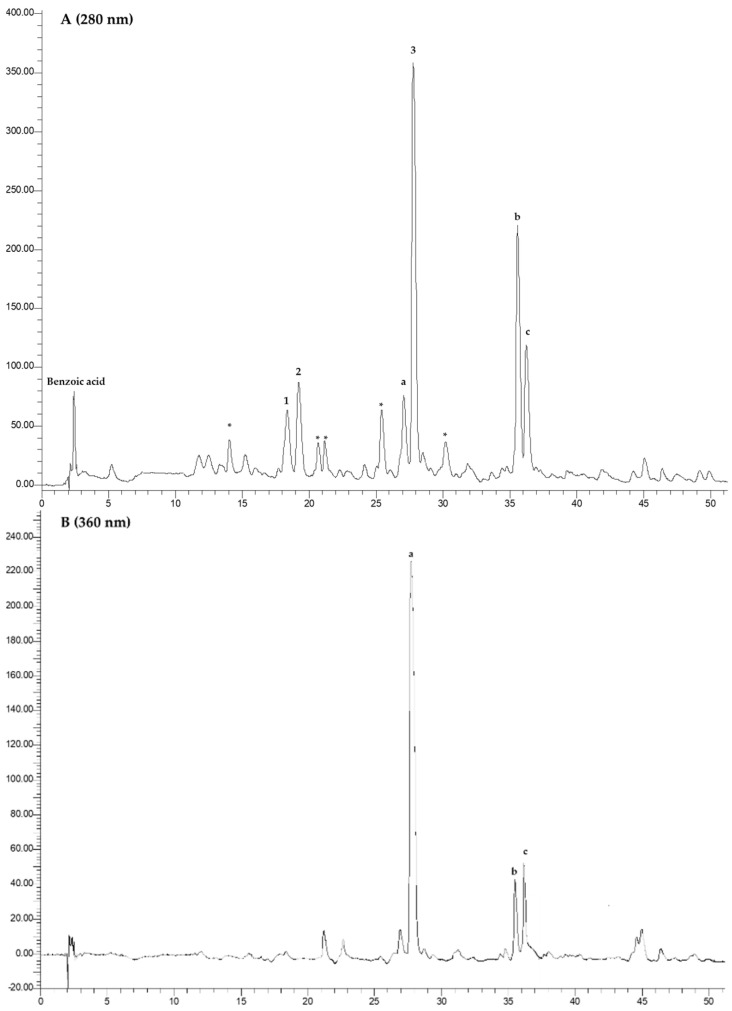
Chromatograms of **R2b** recorded at 280 nm (Panel (**A**)) and at 360 nm (Panel (**B**)). 1. Chlorogenic acid; 2. caffeic acid; 3. sinapic acid; a, b, and c: flavonols; * unknown.

**Figure 11 foods-12-03286-f011:**
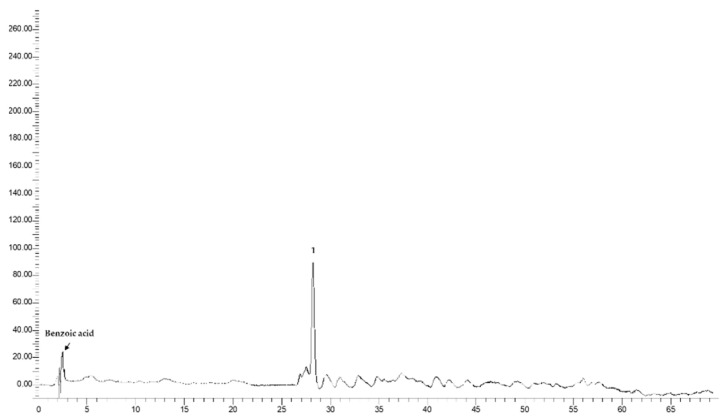
Example chromatograms related to **R3a** and **R3b** recorded at 280 nm. 1. Sinapic acid.

**Figure 12 foods-12-03286-f012:**
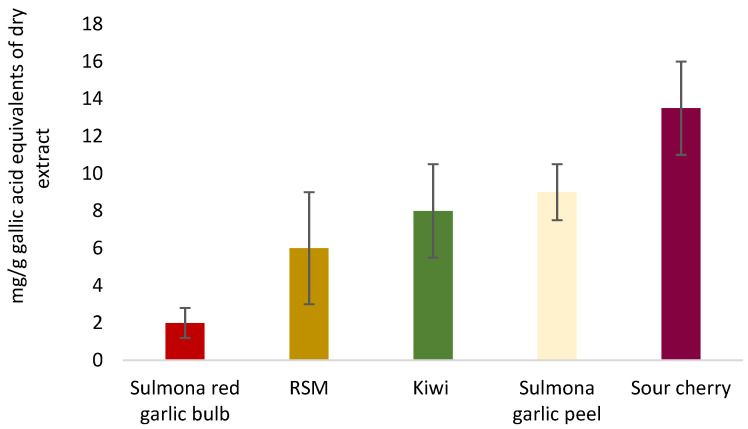
Radical scavenging activities of different food matrices compared with an average value of RSM.

**Table 1 foods-12-03286-t001:** Gravimetric data ^a^.

	Residue	Yield (%)
Ethanol extraction	**R1a**	11.6
	**R2a**	9.0
	**R3a**	35
SFE- CO_2_	**R1b**	0.65
	**R2b**	1.5
	**R3b**	3.6

^a^ Extractions were performed for 5.0 g of each residue.

**Table 2 foods-12-03286-t002:** HPLC-DAD data expressed in µg/g dried extract.

	R1a	R1b	R2a	R2b	R3a	R3b
Benzoic acid ^$^	-	-	-	0.26 ± 0.13	0.52 ± 0.12	0.39 ± 0.16
Chlorogenic acid	-	-	-	0.36 ± 0.19	-	-
Caffeic acid	-	-	-	0.16 ± 0.07	-	-
Sinapic acid	6.59 ± 0.21	-	4.63 ± 0.16	19.38 ± 0.97	4.82 ± 0.21	6.15 ± 0.15
Flavonols *	0.67 ± 0.34	-	-	1.20 ± 0.12	-	-

^$^ expressed as gallic acid equivalents; * expressed as rutin equivalents-not detected.

**Table 3 foods-12-03286-t003:** DPPH data related to analysed samples.

mg/g of Gallic Acid Equivalents of Dry Extract	
**R1a**	2.60 ± 0.27
**R1b**	0.76 ± 0.13
**R2a**	3.24 ± 0.07
**R2b**	6.35 ± 0.58
**R3a**	3.49 ± 0.03
**R3b**	5.42 ± 0.14

## Data Availability

Data are contained within the article or Appendix A.

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
