# Peer review of "Valorisation of Side Stream Products through Green Approaches: The Rapeseed Meal Case"

_foods, 2023, doi:10.3390/foods12173286_

Round 1

Reviewer 1 Report

The manuscript “Valorization of side stream products through green approaches: the rapeseed meal case” is interesting and treats an actual problem (valorisation of agricultural by-products).

This is a really well-written manuscript. The title and abstract are appropriate for the content of the text. The article is well constructed, the experiments were well conducted, and analysis was well performed. I thoroughly enjoyed reviewing this manuscript and I have only one comment for the authors.

The abbreviation of rapeseed meal (RSM) appears in Line 29; please add the full name.

Reviewer 2 Report

Dear authors, 

Thank you for the manuscript! Some comments below - 

In abstract avoid using the abbreaviations R1, R2, R3 - they are not informative and the abstract should be read as a standalone text, so unknown, manuscript-specific abbreaviations are not appropriate.

line 29 - check the reference

line 29 an abbreviation RSM appears but is explained only later, change

line 43, 47 etc - references separated, put in the same bracket

line 70 - the English and the wording is ot really correct, please revise

line 109 - constituted? do you mean consisted of?

line 116 - use pat tense when describing what was done with the samples

line 119 - was it a Restek column? please be precise

in section 2.6. you should also briefly describe the derivatisation procedure, not only mention literature

line 146 - did you really use the polynomial standard curve? there should be a linear relationship berween gallic acid and the response on DPPH. Also it ismore common to use trolox as a standard in this case of DPPH measurements

Table 1 - are those the results of ethanolic extraction or CO2?? write it clearly in the caption

There are way too many NMR spectra, the red letters are unreadable so putting these images have verry little meaning

Table 2 - Benzoic acid (s)?? and in whic extract were these determined? 

What extracts have been measured for DPPH activity?? CO2? Ethanol? You must make this clear

You say that CO2 extracts hsowed highest DPPH activity? How was this masured? was the sample soluble in isopropanol or did you have precipitation in the measurements? what about the lipids in the samples and how they influnce the DPPH assay?

Remove the frame of Figure 3 

in Section 3.2. i see NMR but i dont see any GC analysis? you must show how this lipid-rich material and its extracts look and what they contain using GC!

Where is the discussion part? Please add, put your findings into perspective of previous work!

Reference list must be reworked - there are references in different styles, formatiing and some references dont even have the name of the journal. 2/10.

Overall the article seems a bit rushed. Lacks explanation on most of the things. Also the English must be improved signifficantly.

Must be improved throughout the manuscript.

Reviewer 3 Report

·       The abstract incorporates the word "rapeseed meal" to indicate a connection to the article's title.

·       The meaning of R in the abstract is unclear.

·       Introduction, Line no 11 “rapeseed meal (RSM)” should be stated before “RSM (Line no. 29)”.

·       Line no. 43-44, “Sinapic acid and its esters (sinapin) are the main phenolic acids present in RSM (about 70 %).” When considering the chemical structure of sinapin, it is found that the substance is an alkaloidal amine of sinapic acid. Although sinapin has an ester bond, it is not ester of sinapic acid.

·       It should be present the comparative information on the types and amounts of the active substances as well as the activity remaining in the residues of rapeseed meal obtained from various extraction processes, including conventional ethanol extraction, Soxhlet extraction, and supercritical CO2 extraction.

·       Line no. 63-65, “Although RSM is not considered an excellent source of nutritionally bioactive molecules, it is a rich source of phenolic compounds with nutritional bioavailability and fibres with sensory and functional properties.” This sentence is contradictory and needs to be revised and clarified.

·       Fig.1 “alcaline” should be changed to “alkaline”.

·       In protein removal. it should be described the principle of protein extraction with reference materials.

·       Amino acids are also beneficial for skin health. Why is it necessary to isolate the protein first?

·       Add information on the optimal conditions used in extraction techniques (conventional ethanol extraction, Soxhlet extraction, and supercritical CO2 extraction) ensuring they are suitable conditions and add references.

·       In the introduction, the bioactive substances to be studied by HPLC and GC/MS should be mentioned in order to lead to the analysis of those substances in the experimental section.

·       In order for the data from HPLC and GC/MS to be accurate, precise, and reliable, the method validation of HPLC and GC/MS must be performed.

·       Antioxidant assay requires more than one method, only one method (DPPH) was used in this study.

·       The extracts were dried under vacuum. It should be showed the temperatures.

·       The proton NMR spectrum of the sinapic acid or its derivatives should be shown to compare with the spectrum of the extracts. Because the extract is not a pure substance, signals found at 7-8 ppm may be aromatic protons of substances other than sinapic acid. Methine protons in the yellow band of the structure should have chemical shift values not exceed 6 ppm. Proton NMR integration can help the structure elucidation.

·       HPLC chromatograms of extracts and standards and the values of purity factors obtained from each peak should be shown in this article to confirm that the analyte contents are correct (Table 2).

·       In Fig.3, it may not be able to compare radical scavenging activities of different food matrices from various laboratories due to the variation between laboratories and some modifications in the experiments.

Round 2

Reviewer 2 Report

What are the units used in the spectral figures on Y axis? writ that its intensity.

you still havent corrected the referenc list - for example reference 24 is missing journal/publisher.

English has been improved, some minor corrections only

Author Response

- What are the units used in the spectral figures on Y axis? writ that its intensity.

Thanks for the observation. The units were added.

- You still havent corrected the referenc list - for example reference 24 is missing journal/publisher.

Thanks for the observation. The reference list was corrected.

- English has been improved, with some minor corrections only

Thanks for the observation. Further corrections have been made.

Reviewer 3 Report

·       Described important proton NMR signals on Figures 4-6

·       HPLC chromatograms of extracts and standards and the values of purity factors obtained from each peak should be shown in this article to confirm that the analyte contents are correct (Table 2).

·       Show data of method validation of HPLC-DAD

·       In order for the data from HPLC and GC/MS to be accurate, precise, and reliable, the method validation of HPLC and GC/MS must be performed.

Minor editing

Author Response

- Described important proton NMR signals in Figures 4-6

Thanks for the observation. The description was added.

- HPLC chromatograms of extracts and standards and the values of purity factors obtained from each peak should be shown in this article to confirm that the analyte contents are correct (Table 2).

The chromatograms related to obtained extracts were moved from Supplementary materials and inserted in the text, according to the reviewer's suggestion.

- Show data of method validation of HPLC-DAD.  In order for the data from HPLC and GC/MS to be accurate, precise, and reliable, the method validation of HPLC and GC/MS must be performed.

Related data were added in section 2.8.
